# The Malay-Version Knowledge, Risk Perception, Attitude and Practice Questionnaire on Heatwaves: Development and Construct Validation

**DOI:** 10.3390/ijerph19042279

**Published:** 2022-02-17

**Authors:** Fadly Syah Arsad, Rozita Hod, Norfazilah Ahmad, Mazni Baharom, Fredolin Tangang

**Affiliations:** 1Department of Community Health, Faculty of Medicine, Universiti Kebangsaan Malaysia, Kuala Lumpur 56000, Malaysia; fadlysyaharsad@gmail.com (F.S.A.); norfazilah@ppukm.ukm.edu.my (N.A.); mazni_baharom@yahoo.com (M.B.); 2Department of Earth Sciences and Environment, Faculty of Science and Technology, Universiti Kebangsaan Malaysia, Bangi 43600, Malaysia; tangang@ukm.edu.my

**Keywords:** knowledge, risk perception, attitude, practice, questionnaire, validation, Malay-version

## Abstract

Background: Heatwaves have long been recognised as a serious public health concern. This study was aimed at developing and validating a Malay-version of a questionnaire for evaluating knowledge, risk perception, attitudes, and practices regarding heatwaves. Method: The knowledge construct was evaluated with item analysis and internal reliability. The psychometric characteristics, construct and discriminant validity, and internal consistency of the risk perception, attitude and practice constructs were evaluated with exploratory factor analysis (EFA) and confirmatory factor analysis (CFA). Results: The 16 items in the knowledge construct had a good difficulty, discrimination, and reliability index of 0.81. A total of 16 items were maintained in EFA with Cronbach’s alpha of 0.84 and 0.82, 0.78 and 0.84 obtained for total items and risk perception, attitude, and practice constructs, respectively. A total of 15 items were retained after CFA. The finalised model met the fitness indices threshold. The convergent and discriminant validity were good. Conclusion: This newly developed Malay-version KRPAP questionnaire is reliable and valid for assessing Malaysians’ knowledge, risk perception, attitudes, and practices regarding heatwaves.

## 1. Introduction

Recent reports have stated that climate change will increase extreme weather events, including heatwaves. Heatwaves are characterised by a period of abnormally high temperatures and have been widely recognised as a major public health concern around the world [1,2], consistently linked to increased morbidity and mortality. For example, heatwaves in western Europe in 2003 resulted in approximately 70,000 heat-related fatalities [3]. France, the United States and Russia were among other countries that reported the greatest impact from heatwaves [4]. Heatwaves also increase the risk of heat-related illnesses (HRIs) such as heatstroke, heat rash, heat cramps and heat exhaustion [5]. Heatwaves have an impact on other risks, such as droughts. They have been identified in Europe as a trigger leading to water scarcity and all the secondary risks associated with this domino effect [6].

In 2019, the Malaysian Meteorological Department reported that several states had issued a heatwave alert, including Kedah, Perlis, Perak, Pahang, Kelantan and Sarawak [7]. Although the number of reported cases of heat-related morbidity and mortality in Malaysia is small, it remains a public health concern. This is due to the fact that heatwave events are projected to increase in frequency and intensity in the future, related to climate change [8]. In Malaysia, heatwaves are defined as a daily maximum temperature of 37 °C for three consecutive days [7].

In 2018, 61 cases of heatstroke were reported in Selangor among Malaysian Police Training Centre trainees [9]. With concern about the impact of heatwaves on public health increasing, the authorities, especially the Ministry of Health Malaysia (MOH), have generated several preventive measures and guidelines. For example, the MOH released a Clinical Practice Guideline for managing HRIs soon after the police trainees’ incident.

HRIs are a preventable disease [10]. However, lack of knowledge and awareness of the impact of heatwaves on human health may result in increased risk, especially to the most vulnerable population [10,11]. Early recognition of heatwave events and appropriate adaptive behaviours can prevent HRIs. Several studies have shown that simple behaviour such as drinking sufficient water, reducing outdoor activities and staying in a cool place can reduce the risk of HRIs [12,13,14].

Knowledge of heatwave impact on health is important, especially for recognising the early signs and symptoms of HRIs, HRI risk factors and HRI early preventive measures and treatment. A person’s level of knowledge regarding heatwaves and HRIs will determine their risk perception of this issue. Furthermore, a person’s perception will influence their behaviour, in this case, behaviour regarding heatwave events. According to the Health Belief Model (HBM), people are more likely to take preventive action if they believe that the threat of a health risk is severe, they are personally vulnerable, and if there are less costs than benefits in doing so [15]. Liu et al. (2013) reported that people with better perception will adapt more efficiently to climate change-related events [16]. Adaptation to heatwaves is mainly influenced by the person’s level of knowledge and perception [17].

The questionnaire is one of the most common methods to determine the level of knowledge and perception in a community [18]. A valid questionnaire must have the following characteristics: (i) simplicity and viability; (ii) reliability and precision in wording; (iii) be adequate for the problem being measured; (iv) reflect the underlying theory or concept to be measured; and (v) be capable of measuring change [19]. Several questionnaires have been developed for measuring important domains related to heatwaves, namely knowledge [20,21,22], risk perception [16,20,23,24], attitude [25] and practice [20,26,27]. Two studies [20,23] adopted and adapted previous questionnaires for their item development. However, the majority of the articles mentioned above did not detail the questionnaire’s item generation strategy. On top of this, information on the validation of these questionnaires is limited. For example, the study performed in Australia only reported the reliability result for their questionnaire [20].

Thus, a questionnaire has to undergo a robust process of development and validation, to ensure the credibility of the research findings [18,19]. A valid and reliable psychometric scale is crucial, especially for a local setting, due to sociodemographic and socioeconomic differences, which may influence the level of knowledge, risk perception, attitudes and practices (KRPAP) regarding heatwaves. Information on these domains in Malaysia is scarce. Currently, according to our knowledge, there is no validated questionnaire in the Malay language. This justifies the development and validation of a questionnaire in the local language.

This study aimed at developing and validating a Malay-language questionnaire for measuring KRPAP regarding heatwaves for a Malaysian community. The information gained from this study will help fill the information gap on heatwaves in Malaysia. The information obtained will assist policymakers in planning, preventing and mitigating the impacts of heatwaves on the community.

## 2. Materials and Methods

### 2.1. Study Population

This cross-sectional study was conducted from September 2021 to December 2021. Respondents from the general population were invited to participate virtually, as Malaysia was under lockdown due to the coronavirus disease 2019 (COVID-19) pandemic. The online platforms used were WhatsApp, Facebook and email. These digital platforms were linked to Google Forms. The respondents provided informed consent in the first section of the online survey form, where they were fully informed regarding the intent and purpose of the study. Respondents who agreed to participate clicked an ‘I Agree’ button and submitted their data online. Respondents who did not agree to have their data used in the study clicked an ‘I Do Not Agree’ button and their data were not submitted or collected online. The eligibility criteria were Malaysian nationality and age ≥ 18 years.

### 2.2. Questionnaire Development

#### 2.2.1. Item Generation and Formatting

A comprehensive literature review was conducted to identify published questionnaires [20,21,23,24,25,26,27,28,29,30] related to heatwaves and the relevant measured constructs. HRI-related clinical practice guidelines [31] were also included, especially for the knowledge construct. The preliminary KRPAP questionnaire in English was designed to measure the four main constructs of interest: knowledge, risk perception, attitudes, and practices of the community related to heatwave events. The construct development was guided by theories of HBM and the locus of control.

Items in the knowledge construct were scored as follows: True = 1 point, False = 0 points, Don’t know = 0 points, for positively worded statements [20]. Negatively worded statements were scored as follows: True = 0 points, False = 1 point, Don’t know = 0 points [20]. For the risk perception, attitude and practice constructs, questionnaire scoring was set to a 5-point Likert scale (1 = strongly disagree, 2 = disagree, 3 = unsure, 4 = agree, 5 = strongly agree) [20].

#### 2.2.2. Content Validity and Translation

Content validity was performed to assess the comprehensiveness of the items for measuring the constructs. A total of five panel experts including public health physicians from epidemiology and statistics, environmental health, occupational health, along with climatologists, were involved in this process. Unnecessary items were discarded (refer to Section 2.3.1). Ambiguous questions were rephrased for clarity. The modified items were evaluated for the content validity index (CVI). Each item was scored using the following rating scale: 1 = the item is not relevant to the measured construct, 2 = the item is somewhat relevant to the measured construct, 3 = the item is quite relevant to the measured construct, 4 = the item is highly relevant to the measured construct. As the expert panel consisted of five members, items with item-level CVI (I-CVI) > 0.8 were retained [32].

The preliminary KRPAP questionnaire underwent forward and backward translation by two independent translators: a bilingual researcher and a qualified linguistic expert. These experts helped translate the English items into their Malay versions while preserving the original meaning, fluency and appropriateness of the translated items.

### 2.3. Questionnaire Construct Validation

#### 2.3.1. Pre-Testing and Pilot Testing

Pre-testing was conducted for face validity among 10 adults from the general population to review the clarity and understandability of the items for measuring the target outcomes. A pilot study was conducted among 50 respondents from the general population for assessing the construct validity and reliability of the questionnaire [33]. This process was repeated twice to finalise the translated version of the questionnaire.

For knowledge construct validity, item analysis was performed to assess the difficulty index (*p*) and discriminant index (DI). The *p* was calculated as the percentage of upper and lower 27% of respondents who answered correctly after their obtained scores had been ranked in descending order. The *p* was calculated based on the formula by Gronlund and Linn (1990) [34]. Items with *p* value < 0.2 and >0.9 were removed [35].

The DI was calculated to describe the ability of an item to distinguish between high and low scores (i.e., scores of the upper and lower 27% of respondents after their obtained scores had been ranked in descending order, respectively) [34]. Items with DI < 0.2 were removed [35,36]. Knowledge construct reliability was measured using the Kuder-Richardson Formula 20 reliability index (pKR20) [37]. A pKR20 ≥ 0.6 is considered acceptable [38].

Construct validity testing for risk perception, attitude and practice was performed using exploratory factor analysis (EFA). In EFA, factor extraction was performed using principal component analysis (PCA) and varimax rotation. Factors were retained based on the Kaiser criterion with eigenvalues > 1. Meanwhile, items were retained based on factor loading > 0.4. A Kaiser-Meyer-Olkin (KMO) value > 0.5 indicated adequate sample size, and significant Bartlett’s value < 0.05 was accepted for sphericity valuation testing [39]. The reliability of these constructs was determined by calculating the Cronbach coefficient alpha. A Cronbach coefficient alpha ≥ 0.6 is considered acceptable [38].

#### 2.3.2. Subsequent Validation

Validation was conducted using confirmatory factor analysis (CFA) to verify the factorial structure of the risk perception, attitude and practice constructs identified in the EFA. Sample size was determined using a subject: item ratio of 10:1 [40]. Analyses were performed using all items with factor loading > 0.5 [41]. The model fit was assessed using the maximum likelihood estimate [42]. The models’ goodness of fit was assessed using the following statistics: (i) relative Chi-square < 5 [43], (ii) comparative fit index (CFI) ≥ 0.9 [44], (iii) goodness-of-fit index (GFI) ≥ 0.9 [45], (iv) incremental fit index (IFI) ≥ 0.9 [46], (v) parsimony normed fit index (PNFI) > 0.5 [46] and (v) root mean square error of approximation (RMSEA) ≤ 0.08 [47]. Convergent validity (CV) was assessed by the average variance extracted (AVE) and the composite reliability (CR) of the questionnaire was calculated. CV is accepted if AVE > 0.50 [48] and CR > 0.70 [49]. Discriminant validity is established when the heterotrait to monotrait ratio of correlations (HTMT) value is <0.9 [50].

### 2.4. Statistical Analysis

Statistical analysis was performed using SPSS version 26 and the Amos 24 statistical package (IBM Corp., Armonk, NY, USA). Categorical data are described as the frequency (n) and percentage (%). Normally distributed data are described using the mean and standard deviation (SD).

### 2.5. Ethical Approval

This study received ethical approval from the Universiti Kebangsaan Malaysia (UKM) Research Ethics Committee (UKM PPI/111/8/JEP-2021-091) and the MOH Medical Research and Ethics Committee (NMRR-20-3111-57500).

## 3. Results

### 3.1. Study Population

Table 1 depicts the respondents’ characteristics. This study had a total of 165 respondents. The mean respondent age was 36.3 (SD 8.9) years. Most of the respondents were female (67.5%), of Malay race (69.7%), had tertiary education (92.7%), were employed (60.0%) and were married (50.3%). Most 1were in the lower-income group (43.0%).

### 3.2. Questionnaire Development and Construct Validation

The preliminary questionnaire consisted of 60 items. A total of 47 items with I-CVI > 0.8 were retained after content validity had been determined. For face validity, the respondents reported that the items in the questionnaire were clear and easy to understand, with some minor adjustments.

### 3.3. Pilot Testing

Of the 47 items, 20 measured the knowledge construct. Validity testing using item analysis indicated that four items should be removed due to their poor value (K8, K17, K18, K20) (Table 2). The mean *p* and DI values for the remaining 16 items were 55.68 (SD 15.98) and 0.58 (SD 0.18), respectively. The 16 items in the knowledge construct had a good reliability index (pKR20 = 0.81).

EFA of the risk perception, attitude and practice constructs (27 items) indicated that the KMO index was 0.83. This indicated in turn that the sample size in this validation study was adequate for performing EFA. The initial eigenvalue >1 revealed that three components were consistent with the total constructs developed (total variance explained = 57.77) (Table 3). A total of 11 items were removed due to low factor loading (<0.5). Table 4 shows the findings of the pilot study for risk perception (six items), attitude (four items) and practice (six items). Reliability analysis indicated Cronbach’s alpha of 0.84 and 0.82, 0.78 and 0.84 for total items and the risk perception, attitude and practice constructs, respectively. Appendix A summaries the details of the construct, objectives and corresponding items.

### 3.4. Subsequent Validation

Two models were simulated for the model fit assessment using CFA. The first model showed good fit indices except for the GFI, which was 0.883. Double-headed arrows were used for correlating items with high modification index values and the model fitness improved greatly as in Model 2 (Figure 1). Appendix A summaries the fit indices for both models.

Table 5 shows the convergent validity (AVE) and CR of the three domains. All constructs were valid and reliable for measuring risk perception, attitude and practice regarding heatwaves.

For construct validity and discriminant validity, two models were simulated in this analysis. Some modifications were performed in Model 3 to obtain better AVE, CR, maximum shared variance (MSV) and HTMT warning values (Appendix A). Finally, Model 3 was selected as the final model, where one modification was made (A2 was removed) (Table 5). Figure 2 summarises the overall questionnaire development and validation process.

## 4. Discussion

This Malay-version KRPAP questionnaire on heatwaves is the first validation study performed in Malaysia. In addition, there are currently no available validation studies on heatwaves that measure the construct validity of the items. In the present study, the questionnaire adequately and appropriately covered the intended constructs, i.e., knowledge, risk perception, attitude and practice, based on the HBM.

In terms of psychometric properties, the data obtained in this study were suitable for performing an EFA followed by CFA. The KMO value of 0.83 indicated a good sample [39]. Bartlett’s value of sphericity was significant. In the present study, the total variance was 57.77% which is acceptable for a community survey [51].

In the EFA, the items were grouped into three components consistent with the assigned domains (risk perception, attitude, practice) with good factor loading (>0.5). This showed that the items in each domain appropriately measure the intended objective. However, 16 items were removed in this process. In Malaysia, heatwave events are infrequent and the population occasionally does not even realise that it is facing a heatwave. This may influence the Malaysian population’s perception of the risk of heatwaves, practices for reducing the impact of heatwaves and attitudes for protecting against heatwave impact.

In the CFA, the Malay-version KRPAP questionnaire demonstrated a good fit model. Two models were simulated to achieve good fit indices. Here, Model 1 showed good fit indices except for the GFI. Some modifications were made for Model 2, namely co-variating the selected items to improve the GFI value. These procedures did not violate the model, as the covariates were performed in each domain and there was no domain crossover, which would have been unacceptable.

For construct validity, two models were simulated as Model 2 became the baseline. Model 3 showed better results and was chosen as the final model. In Model 3, domain practice and risk perception had AVE < 0.5. Most validation studies use AVE > 0.5 as an acceptable result [52,53]. However, Sharif et al. (2021) accepted a value of >0.4 [54]. Only one item was removed from Model 2 to achieve an optimum value and at least three items in each domain were preserved in the CFA process [55,56].

Lastly, the Malay-version KRPAP questionnaire demonstrated good overall and subscale internal consistency. However, the internal consistency was lower than that of several previous studies performed outside Malaysia. This may be caused by several factors such as the online distribution of the questionnaire and the number of respondents [57]. In the present study, the only means of distributing the questionnaire was via online platforms. This method was chosen because the COVID-19 pandemic was ongoing during the study. The drawback of the method is that we relied on the respondents to understand the meaning of the items in the questionnaire independently.

## 5. Conclusions

Our development and validation of the Malay language KRPAP questionnaire is the first study in Malaysia to study the community’s knowledge, perception, attitudes and practices regarding heatwaves. The Malay-version KRPAP showed good psychometric properties…. The data obtained were suitable for performing EFA and CFA that demonstrated a factor structure consistent with the domain developed in the initial stage of this study and its theoretical model. Through this study, the validated Malay KRPAP questionnaire can be implemented in the local setting to measure knowledge, perception, attitudes and practices regarding heatwaves. Further studies involving a larger sample population are recommended in the future.

## Figures and Tables

**Figure 1 ijerph-19-02279-f001:**
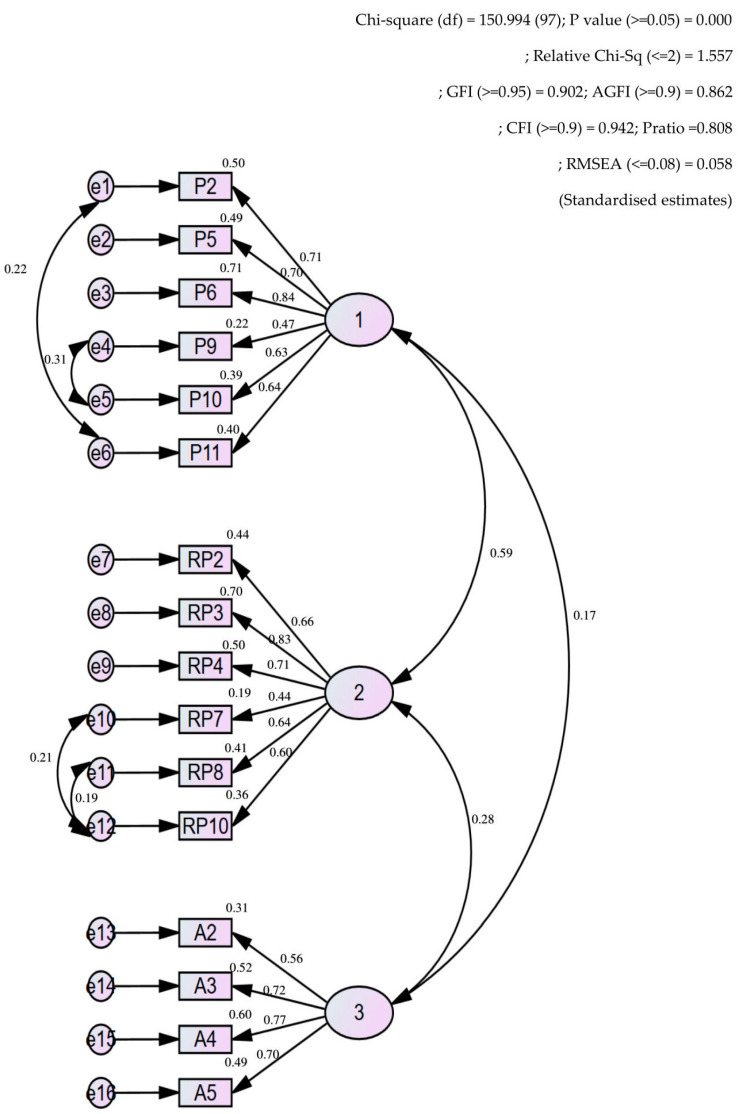
CFA of the risk perception, attitude and practice model.

**Figure 2 ijerph-19-02279-f002:**
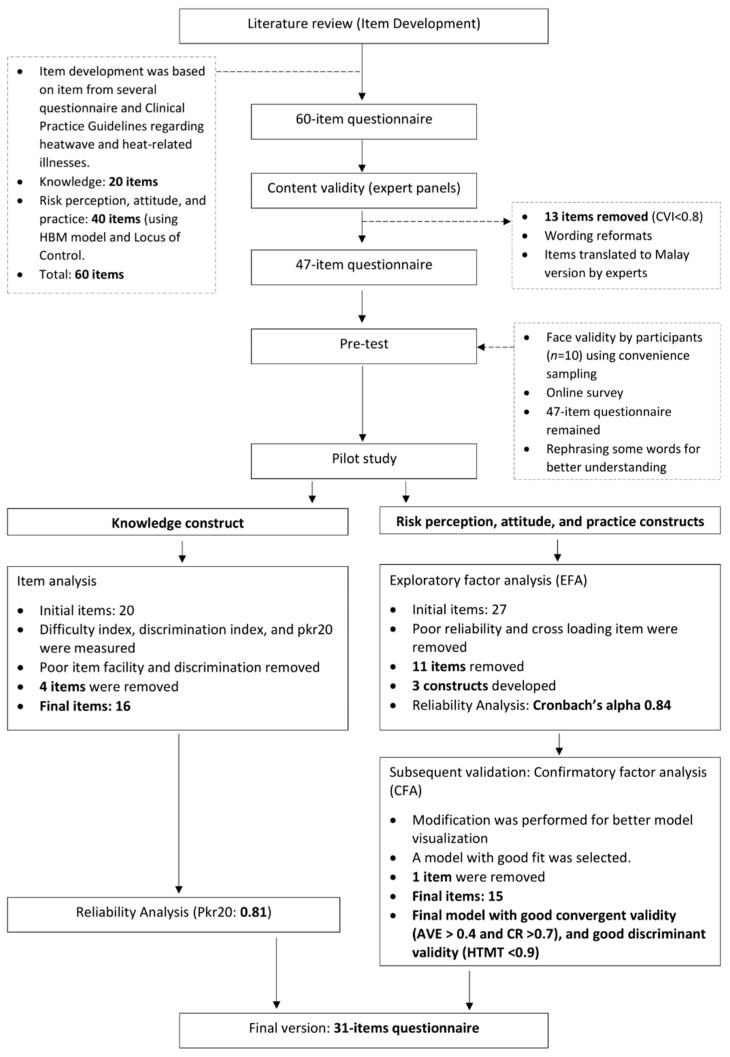
Flowchart of the Malay-version of KRPAP questionnaire development and construct.

**Table 1 ijerph-19-02279-t001:** Characteristics of the respondents (n = 165).

Variables		n (%)
Age (year)	≤20	8 (5)
	21–25	40 (24.2)
	26–30	15 (9)
	31–35	42 (25.5)
	36–40	26 (15.8)
	41–45	14 (8.5)
	46–50	11 (6.7)
	51–55	4 (2.4)
	>55	5 (2.9)
Sex	Male	54 (32.5)
	Female	111 (67.5)
Ethnicity	Malay	115 (69.7)
	Chinese	11 (6.7)
	Indian	8 (4.8)
	Bumiputra Sabah/Sarawak	25 (15.2)
	Others	6 (3.6)
Educational level	No formal school	0 (0.0)
	Primary school	0 (0.0)
	Secondary School	12 (7.3)
	Tertiary education	153 (92.7)
Occupation	Employed	99 (60.0)
	Non-employed	20 (12.1)
	Students	46 (27.9)
Marital status	Married	83 (50.3)
	Divorced/Widowed	3 (1.8)
	Not married	79 (47.9)
Monthly household income	<MYR 4850	71 (43.0)
	MYR 4850–MYR 10,959	60 (36.4)
	>MYR 10,959	34 (20.6)

MYR: Malaysian Ringgit.

**Table 2 ijerph-19-02279-t002:** Validity and reliability analyses for knowledge construct.

Code Item	1st Trial					2nd Trial				
	Difficulty Index (*p*)		Discrimination Index (DI)		Retained Item (√/×)	Difficulty Index (*p*)		Discrimination Index (DI)		Retained Item (√/×)
K1	37.58		0.24		√	37.58		0.29		√
K2	76.97		0.58		√	76.97		0.58		√
K3	54.55		0.64		√	54.55		0.69		√
K4	64.85		0.69		√	64.85		0.71		√
K5	58.79		0.60		√	58.79		0.62		√
K6	21.82		0.24		√	21.82		0.24		√
K7	70.91		0.71		√	70.91		0.67		√
K8	3.64		0.04		×	-		-		-
K9	42.42		0.42		√	42.42		0.36		√
K10	66.67		0.78		√	66.67		0.80		√
K11	50.30		0.56		√	50.30		0.56		√
K12	63.64		0.80		√	63.64		0.78		√
K13	67.88		0.76		√	67.88		0.76		√
K14	25.45		0.38		√	25.45		0.36		√
K15	76.97		0.49		√	76.97		0.53		√
K16	59.39		0.56		√	59.39		0.60		√
K17	16.97		0.04		×	-		-		-
K18	7.88		0.07		×	-		-		-
K19	52.72		0.76		√	52.72		0.76		√
K20	0.61		0.02		×	-		-		-
	Mean 46	SD 24.32	Mean 0.47	SD 0.26		Mean 55.68	SD 15.98	Mean 0.58	SD 0.18	
	pKR20 = 0.78					pKR20 = 0.81				

pKR20: Kuder and Richardson Formula 20 Reliability Index; SD: Standard deviation.

**Table 3 ijerph-19-02279-t003:** Kaiser-Meyer-Olkin and Bartlett’s Test and Total Variance Explained.

Kaiser-Meyer-Olkin Measure of Sampling Adequacy.		0.83
Bartlett’s Test of Sphericity	Approx. Chi-Square	1007.60
	df	120
	Sig.	<0.001
Total Variance Explained (3 components)		57.77

**Table 4 ijerph-19-02279-t004:** Pilot study for risk perception, attitude, and practice constructs.

Construct	Item		Component and Factor Loading			Reliability Analysis		
			1	2	3	Item-Total Correlation	Cronbach’s Alpha	
							(Construct)	(Total)
Risk Perception	RP2			0.74		0.43	0.82	0.84
	RP3			0.76		0.59		
	RP4			0.69		0.56		
	RP7			0.58		0.41		
	RP8			0.59		0.60		
	RP10			0.77		0.49		
Attitude	A2				0.67	0.37	0.78	
	A3				0.78	0.37		
	A4				0.82	0.33		
	A5				0.75	0.36		
Practice	P2		0.76			0.47	0.84	
	P5		0.71			0.47		
	P6		0.77			0.58		
	P9		0.62			0.30		
	P10		0.73			0.50		
	P11		0.72			0.51		

**Table 5 ijerph-19-02279-t005:** Factor loading of each item with their respective construct and the convergent validity, composite reliability measurement.

Construct	Item	Factor Loading	Average Variance Extracted	Composite Reliability	
Risk Perception	RP2	0.74	0.43	0.81	
	RP3	0.75			
	RP4	0.69			
	RP7	0.59			
	RP8	0.60			
	RP10	0.78			
Attitude	A3	0.80	0.54	0.77	
	A4	0.83			
	A5	0.79			
Practice	P2	0.76	0.45	0.82	
	P5	0.72			
	P6	0.77			
	P9	0.61			
	P10	0.73			
	P11	0.73			

## Data Availability

Data is contained within the article or Appendix A.

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
