# Peer review of "The Malay-Version Knowledge, Risk Perception, Attitude and Practice Questionnaire on Heatwaves: Development and Construct Validation"

_ijerph, 2022, doi:10.3390/ijerph19042279_

Round 1
Reviewer 1 Report
The author investigated the ‘The Malay-Version Knowledge, Risk Perception, Attitude and 2 Practice Questionnaire on Heatwaves: Development and Con- 3 struct Validation’using 165 respondents from different backgrounds. Their cross-sectional study was conducted virtually using online softwares between September 2021 and December 2021 and analysed using several statistical methods. The aim of the research which is to develop the Malay version of questionnaire for evaluating knowledge, risk perception, attitudes and practices that are related to heatwaves are adequately and approriately covered by the study.
This is a good research work which the outcome could be used locally to access the knowledge, attitude, perception and practices on heatwaves. Thus, I recommend the manuscript to be publish in your well-organised journal.
Kindly check and treat the underlisted comments.
Line 93, this could be as well be written as: The eligibility criteria are that the participants should be a Malaysian nationality and 18 years of age or above.
Line 111-113, avoid repetitions, re-write sentence as: A total of five panel experts from public health physicians that include epidemiology and statistics, environmental health, occupational health, and climatologist were involved in this process.
Line 177: This is not clear when compared to line 93 - Why is Malay 69.7%, and other people of different ethnicity added in Table 1? I expect 100% in accordance with the eligibility statement in line 93?
Author Response
Dear reviewer,
Please see the attachment.
Thank you.
Warmest regards.

Reviewer 2 Report
Abstract:
General comment: The Abstract is well written. It presents the problem touched upon by the manuscript, the background, the method and the conclusion. However, the manuscript is expected to provide more resolute conclusion, therefore propose to consider if the word “potentially” might be avoided, based on the achieved results.
Introduction:
General comment: Overall the Introduction constitutes a very well set section of the manuscript arguing for the importance of the topic in a referenced based manner, providing proves from a literature review, etc. It also explains limitations and gaps identified in existing validation methods which are to be covered by this paper. The Introduction properly identifies and informs about the expected impact of the paper as well as its target group. A research question to be answered with this article is slightly missing. It is recognized as a minor shortcoming to be amended.
Line 36-37: Heatwaves influence also other risks e.g. such as droughts. They have been recognised in Europe in Danube River Basin as a trigger leading to water scarcity and all secondary risks related to this domino effect.
Material and Methods:
Line 103-105: Scoring of the responses needs further argumentation, reasoning and explanation. Providing references to a literature sources from which such scores are rooted. Referencing to literature sources might be sufficient.
Line 114: “Unnecessary items were discarded” – please provide examples of „unnecessary items” or refer them to the section 2.3.1.
Results:
General comment: Preferably, please present the final items (questions) that are included in the accepted Model 2 of the questionnaire. It would enrich the content, make it more grounded and easy up understanding of the study subject. As a matter of fact the model is the key result of the study.
Line 202: Suggest to not refer to supplementary files as they will not be published, so a reader might be confused how he/she can get access to it.
Line 202: comment as above.
Line 217: comment as above.
Table 5: it is not clear why “Risk Perception” is bolded…
Discussion:
General comment: Please discuss broader the shortcomings and strengths of the two researched models providing examples of items (questions).
Line 245-256: please make it fully clear what you meant by the term model 1 and model 2 in the text. It is not fully clear.
Author Response

(The authors gave the same response as above.)
